# CLUSTERING FOR PROTEIN REPRESENTATION LEARNING

## ABSTRACT

Protein representation learning is a challenging task that aims to capture the structure and function of proteins from their amino acid sequences. Previous methods largely ignored the fact that not all amino acids are equally important for protein folding and activity. In this article, we propose a neural clustering framework that can automatically discover the critical components of a protein by considering both its primary and tertiary structure information. Our framework treats a protein as a graph, where each node represents an amino acid and each edge represents a spatial or sequential connection between amino acids. We then apply an iterative clustering strategy to group the nodes into clusters based on their 1D and 3D positions and assign scores to each cluster. We select the highest-scoring clusters and use their medoid nodes for the next iteration of clustering, until we obtain a hierarchical and informative representation of the protein. We evaluate our framework on four protein-related tasks: protein fold classification, enzyme reaction classification, gene ontology term prediction, and enzyme commission number prediction. Experimental results demonstrate that our method achieves state-of-the-art performance. Our code will be released.

## 1 INTRODUCTION

Proteins are one of the most fundamental elements in living organisms and make significant contributions to nearly all fundamental biological processes in the cell. Composed of one or several chains of amino acids [1, 2], proteins fold into specific conformations to enable various biological functionalities [3]. A multi-level structure of proteins begins with the *primary structure*, which is defined by the sequence of amino acids forming the protein backbone [4]. The *secondary structure* is determined by hydrogen bonds between distant amino acids in the chain, resulting in substructures such as $\alpha$-helices and $\beta$-sheets [5]. *Tertiary structure* arises from folding of secondary structures, determined by interactions between side chains of amino acids [6]. Lastly, the *quarternary structure* describes the arrangement of polypeptide chains in a multi-subunit arrangement [7]. Understanding the structure and function of proteins [8–12] is crucial in elucidating their role in biological processes and developing new therapies and treatments for a variety of diseases.

While a protein's conformation and function are primarily determined by its amino acid sequence, it is important to recognize that not all amino acids contribute equally to these aspects. In fact, certain amino acids, known as the *critical components*, play the primary role in determining a protein's shape and function [13–19]. Sometimes, even a single amino acid substitution can significantly impact a protein's overall structure and function [13, 14]. For example, sickle cell anemia results from a single amino acid change in hemoglobin, causing it to form abnormal fibers that distort red blood cell shape. Besides, the critical components of a protein's primary structure are essential for its biological activity. For instance, any of the first 24 amino acids of the adrenocorticotropic hormone (ACTH) molecule is necessary for its biological activity, whereas removing all amino acids between 25-39 has no impact [15, 16]. Also, proteins with the same critical components perform the same function, *e.g.*, the insulin A and B chains in various mammals contain 24 invariant amino acid residues necessary for insulin function, while differences in the remaining amino acid residues do not impact insulin function [17, 18]. In addition, proteins from the same family often have long stretches of similar amino acid sequences within their primary structure [19], suggesting that only a small portion of amino acids that differentiate these proteins are the critical components.

Motivated by the fact that certain amino acids play a more critical role in determining a protein's structure and function than the others [13–19], we devise a neural clustering framework for protein

representation learning. Concretely, it progressively groups amino acids so as to find the most representative ones for protein classification. During each iteration, our algorithm proceeds three steps: spherosome cluster initialization (SCI), cluster representation extraction (CRE), and cluster nomination (CN). The iterative procedure starts by treating a protein as a graph where each node represents an amino acid, and each edge represents a spatial or sequential connection between amino acids. In SCI step ($\triangle$), all the nodes are grouped into clusters based on their sequential and spatial positions. Subsequently, in CRE step ($\square$), the information of neighboring nodes within the same cluster are aggregated into a single representative node, namely medoid node. This step effectively creates an informative and compact representation for each cluster. Lastly, in CN step ($\triangledown$), a graph convolutional network (GCN) [20] is applied to score all the clusters, and a few top-scoring ones are selected and their medoid nodes are used as the input for the next iteration. By iterating these steps $\circlearrowright(\triangle\square\triangledown)$, we explore the protein's structure and discover the representative amino acids for protein representation, leading to a powerful, neural clustering based protein representation learning framework.

By embracing the powerful idea of clustering, our approach favorably outperforms advanced competitors. We observe notable improvements of **5.6**% and **2.9**% in $F_{\max}$ for enzyme commission number prediction [8] (§4.1) and gene ontology term prediction [8] (§4.2). Our method also yields remarkable enhancements of **3.3**% and **1.1**% in classification accuracy for protein fold classification [21] (§4.3) and enzyme reaction classification [9] (§4.4). We also provide comprehensive diagnostic analyses (§4.5) and visual results (§4.6), verifying the efficacy of our essential algorithm designs, showing strong empirical evidence for our core motivation, and confirming the capability of our algorithm in identifying functional motifs of proteins. We will make our code publicly available.

## 2 RELATED WORK

**Protein Representation Learning**. It has been a topic of interest in the field of bioinformatics and computational biology in recent years. Existing methods for this topic can be broadly categorized into two types: *sequence-based* and *structure-based*. Early works on sequence-based protein representation learning typically apply word embedding algorithms [22, 23] and 1D convolutional neural networks [21, 24–26]. Though straightforward, these methods neglect the spatial information in protein structures. To address this limitation, structure-based methods explore the use of 3D convolutional neural networks [27–29] and GCNs [8, 10, 11, 30–33] for this task. Recently, some approaches [11, 33] focus on atom-level representations, treating each atom as a node. The state-of-the-art performance has been achieved by learning at the amino acid-level [10, 12], indicating that protein representation is more closely related to amino acids rather than individual atoms.

Despite significant progress made by these existing methods, they treat all amino acids equally. In sharp contrast, we propose to learn the protein representation by a neural clustering framework. This allows us to capture the inherent variability and significance of different amino acids, leading to a more comprehensive and accurate representation of proteins. We are of the opinion that our method has several potential applications and extensions in the field of protein science. For instance, our neural clustering approach can benefit protein design by utilizing the learned crucial components to direct the design of novel protein sequences [34, 35] that possess specific properties or functions. This, in turn, can facilitate the development of new therapies and treatments for a diverse range of illnesses.

**Clustering.** Clustering is a fundamental data analysis task that aims to group similar samples together based on similarity, density, intervals or particular statistical distribution measures of the data space [36–38]. It helps to identify representative patterns in data which is meaningful for exploratory data analysis. Traditional clustering methods [39, 40] heavily rely on the original data representations. As a result, they often prove to be ineffective when confronted with data residing in high-dimensional spaces, such as images and text documents. Recently, deep learning-based clustering methods have attracted increased attention, and been successfully applied to various real-world applications, such as image segmentation [41, 42], unsupervised representation learning [43, 44], financial analysis [45–47], and text analysis [48–50].

Drawing inspiration from the biological fact that the significance of various amino acids varies, we propose a neural clustering framework for end-to-end protein representation learning. Our objective is to leverage the inherent benefits of clustering to identify the representative amino acids. In experiments, we demonstrate the feasibility of our clustering-based method through numerical evaluations and provide visual evidence to reinforce the underlying motivation behind our approach.

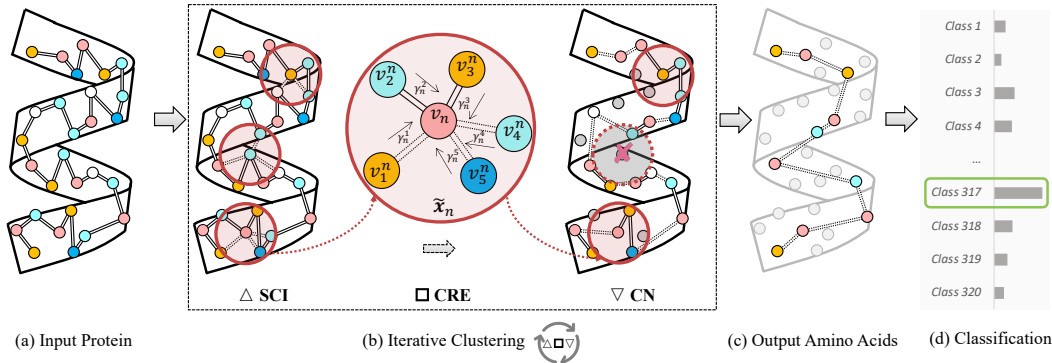

(a) Input Protein  (b) Iterative Clustering  (c) Output Amino Acids  (d) Classification

Figure 1: Overview of our iterative neural clustering pipeline for protein representation learning: (a) input protein with amino acids, (b) iterative clustering algorithm which repeatedly stacks three steps $\circlearrowright(\triangle\square\triangledown)$, (c) output can be seen as the critical amino acids of the protein, (d) output amino acids used for classification. The details of our iterative neural clustering method can be seen in §3.2.

## 3 METHODOLOGY

We have stated that the structure and function of a protein are represented by certain critical amino acids in §1. This motivates us to regard the protein representation learning task as an amino acid clustering process, so as to automatically explore the critical components of the protein. Firstly, some basic concepts used throughout the paper and the task setup are introduced in §3.1. Then, we elaborate on the neural clustering framework in §3.2. Finally, §3.3 presents our implementation details.

### 3.1 NOTATION AND TASK SETUP

The problem of protein representation learning can be formalized as follows. A protein is denoted as a triplet $\mathcal{P} = (\mathcal{V}, \mathcal{E}, \mathcal{Y})$, where $\mathcal{V} = \{v_1, \cdots, v_N\}$ is the set of nodes representing $N$ amino acids, $\mathcal{E}$ the set of edges representing spatial or sequential connections between amino acids, and $\mathcal{Y}$ the set of labels. The target goal of protein classification is to learn a mapping $\mathcal{V} \to \mathcal{Y}$. Specifically, in single-label classification, *e.g.*, protein fold classification and enzyme reaction classification, the focus is on learning from a collection of examples that are linked to a *single* label from $\mathcal{Y}$. While in multi-class classification, *e.g.*, enzyme commission number prediction and gene ontology term prediction, each examples are associated with *multiple* labels from $\mathcal{Y}$. In what follows, we use $\{x_1, \cdots, x_N\}$ to denote the features of $\mathcal{V}$, where $x_n \in \mathbb{R}^{256}$ is the feature vector of amino acid $v_n$. The feature vector can be derived from various sources, such as the one-hot encoding of amino acid types, the orientations of amino acids, and the sequential and spatial positions of amino acids. We use $A \in \{0, 1\}^{N \times N}$ to denote the adjacency matrix of $\mathcal{V}$, where $A_{n,m} = 1$ if there exists an edge between amino acid nodes $v_n$ and $v_m$, *i.e.*, $e_{n,m} \in \mathcal{E}$.

### 3.2 ITERATIVE CLUSTERING

In our neural clustering framework, we perform iterative clustering on amino acids of the input protein. Each iteration encompasses three steps, Spherosome Cluster Initialization (SCI), Cluster Representation Extraction (CRE), and Cluster Nomination (CN), as illustrated in Fig. 1.

**Spherosome Cluster Initialization (SCI).** For each amino acid, we initialize a cluster by considering other locally neighboring amino acids within a fixed receptive field. This approach enables the examination and comprehension of the spatial and sequential associations among amino acids, which hold paramount significance in determining the structure and functioning of proteins []. Specifically, for each amino acid $v_n$, we define a cluster as a set of amino acids within a fixed radius $r$, where $v_n$ is regarded as the *medoid* node of the cluster. The fixed radius $r$ is a hyperparameter which determines the extent to which the local amino acid nodes are considered for cluster initialization, and its impact on the final performance of protein analysis is studied in §4.5. For $v_n$, we denote the set of its neighbor amino acid nodes as $\mathcal{H}^n = \{v_1^n, \cdots, v_K^n\}$. Note that $v_n \in \mathcal{H}^n$. In the first iteration ($t=1$), we conduct SCI process based on all the input $N$ amino acids to form the clusters. In subsequent iterations ($t > 1$), we use the nominated $N_{t-1}$ amino acids from the previous $t-1$-th iteration to initialize the clusters. This allows to focus on exploring the critical components of the protein graph and avoid redundantly exploring the same areas. The adjacency matrix $A$ is regenerated in each SCI process with considering the connectivity among amino acids.

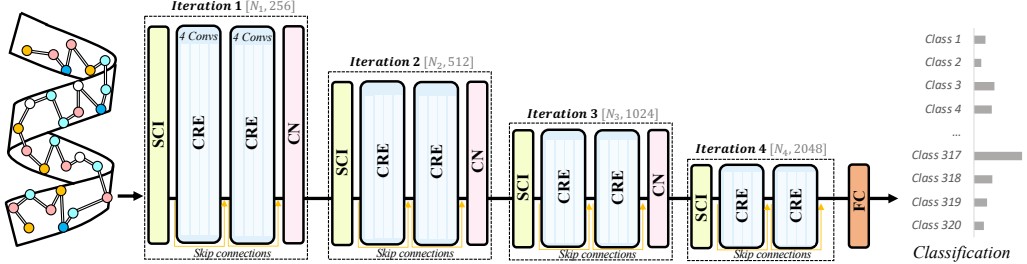

Figure 2: Our neural clustering framework architecture with four iterations. Given a protein, a set of 1D and 3D amino acids, our method adopts an iterative clustering algorithm to explore the most representative amino acids. At each iteration, $B$ cluster representation extraction blocks are utilized to extract cluster features. The clustering nomination operation selects the fraction $\omega$ of amino acids for the next iteration, that $N_t = \lfloor \omega \cdot N_{t-1} \rfloor$. Details of the framework can be seen in §3.3.

**Cluster Representation Extraction (CRE).** The second step aims to learn the overall representation of each cluster $\mathcal{H}^n$ by considering all the amino acids within it. Specifically, we construct the feature representation $\boldsymbol{x}_k^n$ of the amino acid node $v_k^n$ in the cluster $\mathcal{H}^n$ by:

$$\boldsymbol{x}_k^n = f(\boldsymbol{g}_k^n, \boldsymbol{o}_k^n, d_k^n, s_k, \boldsymbol{e}_k), \tag{1}$$

where $\boldsymbol{g}_k^n = (\boldsymbol{z}_k - \boldsymbol{z}_n) \in \mathbb{R}^3$ denotes relative geometric coordinates, and $\boldsymbol{z}$ indicates the spatial coordinate; $\boldsymbol{o}_k^n \in \mathbb{R}^3$ is the 3D orientation vector; $d_k^n = (||\boldsymbol{z}_k - \boldsymbol{z}_n||_2) \in \mathbb{R}^3$ indicates the spatial distance between $v_n$ and $v_k^n$; $s_k$ is the sequential order on the amino acid chain that is relative to the beginning of the protein sequence; $\boldsymbol{e}_k \in \mathbb{R}^{128}$ denotes amino acid embedding (*i.e.*, one-hot encoding of the amino acid type) of $v_k^n$ in the cluster; and $f$ denotes an encoder which is implemented by a multilayer perceptron. In this way, our neural clustering framework considers both primary (1D sequence-based distance) and tertiary (3D coordinates and orientations) structure information of proteins. Then, we utilize the cross-attention mechanism [51] to calculate the attention score $\gamma_k^n \in (0, 1)$ between the medoid amino acid node feature $\boldsymbol{x}_n$ and all the constituent ones $\boldsymbol{x}_k^n$ in the cluster $\mathcal{H}^n$:

$$\gamma_k^n = \frac{\exp(\boldsymbol{w}[\boldsymbol{x}_n, \boldsymbol{x}_k^n])}{\sum_{k=1}^K \exp(\boldsymbol{w}[\boldsymbol{x}_n, \boldsymbol{x}_k^n])}, \tag{2}$$

where $\boldsymbol{w}$ is a learnable vector, and $[\cdot]$ refers to the concatenation operation. The attention score $\gamma_k^n$ denotes the level of focus of the medoid on other constituent amino acids. Then, the overall cluster representation $\tilde{\boldsymbol{x}}_n$ for the cluster $\mathcal{H}^n$ of node $v_n$ is given as:

$$\tilde{\boldsymbol{x}}_n = \sum_{k=1}^K \gamma_k^n \boldsymbol{x}_k^n. \tag{3}$$

**Cluster Nomination (CN).** To identify the most representative amino acids (*i.e.*, critical components) in the protein, we propose a cluster nomination process that learns to automatically select these critical components based on the cluster representations $\{\tilde{\boldsymbol{x}}_n\}_{n=1}^N$. Specifically, such cluster nomination process is achieved by a GCN, which takes each cluster representation $\tilde{\boldsymbol{x}}_n$ as input and calculates a *nomination* score $\varphi_n \in (0, 1)$:

$$\varphi_n = \sigma(\boldsymbol{W}_1 \tilde{\boldsymbol{x}}_n + \sum_{m=1}^{N_t} A_{n,m}(\boldsymbol{W}_2 \tilde{\boldsymbol{x}}_n - \boldsymbol{W}_3 \tilde{\boldsymbol{x}}_m)), \tag{4}$$

where $\boldsymbol{W}_{1,2,3}$ are learnable parameters and $\sigma$ is ReLU function. By utilizing self-loops and the capability to learn functions of local extremas, we are able to score clusters based on both their global and local significance. The cluster feature is then weighted by the calculated scores:

$$\hat{X}^c = \Phi \odot X^c, \tag{5}$$

where $\Phi = [\varphi_1, \cdots, \varphi_{N_t}]^\top$, $X^c = [\tilde{\boldsymbol{x}}_1, \cdots, \tilde{\boldsymbol{x}}_{N_t}]$, and $\odot$ is broadcasted hadamard product. Based on the weighted cluster features $\hat{X}^c$ and the calculated nomination scores $\Phi$, we select top-$N_t$ clusters at the $t$-th iteration. Then, the top-$N_t$ amino acids are utilized to form a new graph, viewed as the input for the next $t+1$-th iteration. Here $N_t$ is the number of selected clusters at $t$-th iteration and is determined by the cluster nomination fraction $\omega$, denoted as $N_t = \lfloor \omega \cdot N_{t-1} \rfloor$, that $t \in \{0, ..., T\}$ and $N_0 = N$. We will illustrate the impact of different values of $\omega$ in §4.5.

At each clustering iteration, we estimate cluster membership and centers by considering the sequential and spatial information of amino acids. Thus it explicitly probes the structures of the portion

and captures the complex relationships among amino acids. Then, the representative amino acids (*i.e.*, cluster medoids) are identified and only those most informative ones (which can be viewed as critical components) are selected for next-iteration clustering, and eventually used for functionality prediction. After $T$ iterations, a fully connected layer is used to project the feature representations of the nominated $N_T$ amino acids to a $|\mathcal{Y}|$-dimension score vector for classification. Note that the attention-based selection is implicitly learnt from the supervision signals of the protein classification task. In a nutshell, our method utilizes an iterative clustering algorithm that repeatedly stacks three steps: SCI, CRE, and CN. The pseudo code of our algorithm can be found in the appendix.

## 3.3 IMPLEMENTATION DETAILS

**Network Architecture.** We progressively nominate $N_T$ from $N$ amino acids of the protein by $T$ iterations (see Fig. 2). We empirically set $T = 4$, suggesting the neural clustering framework consists of four iterations. At $t$-th iteration, we stack $B = 2$ CRE blocks to learn the representation of the selected $N_t$ amino acids. In this way, our method is also a hierarchical framework that downsamples amino acids as the iteration proceeds. In order to ensure a large enough receptive field at late iterations, we increase the value of the cluster radius $r$ applied in SCI step as the number of iterations increases. Concretely, the radii for the four iterations are $r$, $2r$, $3r$, and $4r$, respectively. We adopt skip connection [52] per CRE block to facilitate information flow and ease network training. Inspired by [12, 53, 54], we adopt rotation invariance (detailed in the appendix).

**Training Objective.** We follow the common protocol in this field [10, 12, 54, 55] to append a fully connected neural network as the classification head at the tail of network. Softmax activation and cross-entropy loss are used for single-label classification, while Sigmoid function and binary cross-entropy loss for multi-label classification.

**Reproducibility.** We implement our method using PyTorch-Geometric library. For all our experiments, the training and testing are conducted on a single Nvidia RTX 3090 GPU with 24 GB memory and Intel(R) Xeon(R) Gold 6326 CPU@2.90GHz. More details of the training setup are given in the appendix. To ensure reproducibility, our full implementations and models will be released.

## 4 EXPERIMENTS

We evaluate our method on four tasks following previous studies [9, 10, 12]: enzyme commission (EC) number prediction (§ 4.1), gene ontology (GO) term prediction (§ 4.2), protein fold classification (§ 4.3), and enzyme reaction classification (§ 4.4). Then, in § 4.5, we present a series of diagnostic studies. Finally, we provide a set of visual results in § 4.6 for in-depth analysis. More experimental results and implementation details are provided in the appendix.

### 4.1 EXPERIMENT ON EC NUMBER PREDICTION

**Task and Dataset.** Enzyme Commission (EC) number prediction seeks to anticipate the EC numbers of diverse proteins that elucidate their role in catalyzing biochemical reactions. The EC numbers are chosen from the third and fourth levels of the EC tree, resulting in $538$ distinct binary classification tasks. As in [8], the experimental dataset of this task consists of $15,550/1,729/1,919$ proteins in `train`/`val`/`test` set, respectively. For GO term and EC number prediction, we follow the multi-cutoff splits in [8] to ensure that the `test` set only contains PDB chains with a sequence identity of no more than 95% to the proteins in the `train` set.

**Training Setup and Evaluation Metric.** EC number prediction can be regarded as a multi-label classification task. The performance is evaluated by the protein-centric maximum F-score $F_{\max}$, which is based on the precision and recall of the predictions for each protein.

**Performance Comparison.** We compare our neural clustering method with 11 top-leading methods in Table 1. As seen, our method establishes a new state-of-the-art on EC number prediction task. It surpasses CDConv [12] by **5.6**% ($0.820 \rightarrow \mathbf{0.866}$) and GearNet [10] by **6.9**% ($0.810 \rightarrow \mathbf{0.866}$) in terms of $F_{\max}$. This indicates that our method can learn informative representations of proteins that reflect their functional roles in catalyzing biochemical reactions.

### 4.2 EXPERIMENT ON GO TERM PREDICTION

**Task and Dataset.** GO term prediction aims to forecast whether a protein belongs to certain GO terms. These terms categorize proteins into functional classes that are hierarchically related and organized into three sub-tasks [8]: molecular function (MF) term prediction consisting of 489 classes, biological

Table 1: $F_{max}$ on EC and GO prediction and Accuracy (%) on fold and reaction classification. [†] denotes results taken from [55] and [*] denotes results taken from [9] and [56]. (§4.1-§4.4)

| Method | Publication | EC | GO | | | Fold Classification | | | | Reaction |
|---|---|---|---|---|---|---|---|---|---|---|
| | | | BP | MF | CC | Fold | Super. | Fam. | Avg. | |
| ResNet [57] | *NeurIPS 2019* | 0.605 | 0.280 | 0.405 | 0.304 | 10.1 | 7.21 | 23.5 | 13.6 | 24.1 |
| LSTM [57] | *NeurIPS 2019* | 0.425 | 0.225 | 0.321 | 0.283 | 6.41 | 4.33 | 18.1 | 9.61 | 11.0 |
| Transformer [57] | *NeurIPS 2019* | 0.238 | 0.264 | 0.211 | 0.405 | 9.22 | 8.81 | 40.4 | 19.4 | 26.6 |
| GCN [58] | *ICLR 2017* | 0.320 | 0.252 | 0.195 | 0.329 | 16.8* | 21.3* | 82.8* | 40.3* | 67.3* |
| GAT [59] | *ICLR 2018* | 0.368 | $0.284^{†}$ | $0.317^{†}$ | $0.385^{†}$ | 12.4 | 16.5 | 72.7 | 33.8 | 55.6 |
| GVP [33] | *ICLR 2021* | 0.489 | $0.326^{†}$ | $0.426^{†}$ | $0.420^{†}$ | 16.0 | 22.5 | 83.8 | 40.7 | 65.5 |
| 3DCNN [28] | *Bioinform 2018* | 0.077 | 0.240 | 0.147 | 0.305 | 31.6* | 45.4* | 92.5* | 56.5* | 72.2* |
| GraphQA [32] | *Bioinform 2021* | 0.509 | 0.308 | 0.329 | 0.413 | 23.7* | 32.5* | 84.4* | 46.9* | 60.8* |
| New IEConv [56] | *ICLR 2022* | 0.735 | 0.374 | 0.544 | 0.444 | 47.6* | 70.2* | 99.2* | 72.3* | 87.2* |
| GearNet [10] | *ICLR 2023* | 0.810 | 0.400 | 0.581 | 0.430 | 48.3 | 70.3 | 99.5 | 72.7 | 85.3 |
| ProNet [11] | *ICLR 2023* | - | - | - | - | 52.7 | 70.3 | 99.3 | 74.1 | 86.4 |
| CDConv [12] | *ICLR 2023* | 0.820 | 0.453 | 0.654 | 0.479 | 56.7 | 77.7 | 99.6 | 78.0 | 88.5 |
| Ours | - | **0.866** | **0.474** | **0.675** | **0.483** | **63.1** | **81.2** | **99.6** | **81.3** | **89.6** |

process (BP) term prediction including $1,943$ classes, cellular component (CC) term prediction with $320$ classes. The dataset contains $29,898/3,322/3,415$ proteins for `train`/`val`/`test`, respectively.

**Training Setup and Evaluation Metric.** GO term prediction is also a multi-label classification task. The protein-centric maximum F-score $F_{max}$ is reported.

**Performance Comparison.** We compare our method with 12 existing state-of-the-art methods for protein representation learning on the task of predicting the GO term of proteins, where most of them are CNN or GNN-based methods. The results are shown in Table 1, where our framework achieves competitive $F_{max}$ scores on all three sub-tasks, especially on MF (**0.675** *vs* 0.654) and BP (**0.474** *vs* 0.453) terms, compared to CDConv [12]. Also, our method is clearly ahead of the second-best method, GearNet [10], by large margins, *i.e.*, **0.474** *vs* 0.400 BP, **0.675** *vs* 0.581 MF, **0.483** *vs* 0.430 CC. Our new records across the three sub-tasks show that our neural clustering method can learn rich representations of proteins that capture their functional diversity.

## 4.3 EXPERIMENT ON PROTEIN FOLD CLASSIFICATION

**Task and Dataset.** Protein fold classification, firstly introduced in [21], aims to predict the fold class label of a protein. It contains three different evaluation scenarios: 1) Fold, where proteins belonging to the same superfamily are excluded during training, $12,312/736/718$ proteins for train/-val/ test, 2) Superfamily, where proteins from the same family are not included during training, $12,312/736/1,254$ proteins for `train`/`val`/`test`, 3) Family, where proteins from the same family are used during training, $12,312/736/1,272$ proteins for `train`/`val`/`test`.

**Training Setup and Evaluation Metric.** Protein fold classification can be seen as a single-label classification task. Mean accuracy is used as the evaluation metric.

**Performance Comparison.** In Table 1, we continue to compare our framework with these state-of-the-art methods on the task of classifying proteins into different fold classes. The fold class describes the overall shape and topology of a protein. Our framework yields superior performance. For example, it yields superior results as compared to CDConv [12] by **6.4**%, ProNet [55] by **10.4**% and GearNet [10] by **14.8**% on the Fold evaluation scenario. Considering that protein fold classification is challenging, such improvements are particularly impressive. Across the board, our neural clustering framework surpasses all other methods of protein fold classification, demonstrating that our framework can learn robust representations of proteins that reflect their structural similarity.

## 4.4 EXPERIMENT ON ENZYME REACTION CLASSIFICATION

**Task and Dataset.** Enzyme reaction classification endeavors to predict the enzyme-catalyzed reaction class of a protein, utilizing all four levels of the EC number to portray reaction class. We utilize the dataset processed by [9], which consists of 384 four-level EC classes and $29,215/2,562/5,651$ proteins for `train`/`val`/`test`, where proteins have less than 50% sequence similarity in-between splits.

**Training Setup and Evaluation Metric.** Enzyme reaction classification is regarded as a single-label classification task. We adopt Mean accuracy as the evaluation metric.

**Performance Comparison.** Table 1 presents comparison results of classifying proteins into different enzyme reaction classes. In terms of classification accuracy, our neural clustering framework outperforms the classic GCN-based method by a margin, *e.g.*, GCN [58] by **22.3**%, GAT [59] by **34**%, and GrahQA [32] by **28.8**%. In addition, it surpasses recent three competitors, *i.e.*, CDConv [12] (**+1.1**%), ProNet [55] (**+3.2**%), and GearNet [10] (**+4.3**%). In summary, the proposed neural cluster-

Table 2: Ablative experiments for the neural clustering algorithm. (a) An off-the-shelf clustering algorithm; (b) A simple average pooling method; (c) Randomly generate attention score $\gamma_k^n$(§4.5).

| Method | EC | GO | | | Fold Classification | | | | Reaction |
| | | BP | MF | CC | Fold | Super. | Fam. | Avg. | |
| --- | --- | --- | --- | --- | --- | --- | --- | --- | --- |
| (a) | 0.792 | 0.385 | 0.579 | 0.429 | 43.1 | 67.1 | 99.1 | 69.8 | 86.8 |
| (b) | 0.817 | 0.452 | 0.641 | 0.453 | 57.2 | 78.7 | 99.3 | 78.4 | 88.1 |
| (c) | 0.765 | 0.342 | 0.567 | 0.415 | 44.6 | 69.5 | 99.2 | 71.1 | 86.4 |
| Ours | **0.866** | **0.474** | **0.675** | **0.483** | **63.1** | **81.2** | **99.6** | **81.3** | **89.6** |

Table 3: Efficiency comparison to SOTA competitors on enzyme reaction classification (§4.5).

| Method | Acc. | Runing Time |
| --- | --- | --- |
| New IEConv [56] | 87.2% | 75.3 ms |
| GearNet [10] | 85.3% | *OOM* |
| ProNet [11] | 86.4% | 27.5 ms |
| CDConV [12] | 88.5% | 10.5 ms |
| Ours | 89.6% | 10.9 ms |

ing framework achieves outstanding performance against state-of-the-art methods, suggesting that our method learns informative representations of proteins that reflect their catalytic activity.

### 4.5 DIAGNOSE ANALYSIS

**Neural Clustering.** To demonstrate the effectiveness of neural clustering, we compare it against three baseline approaches that employ naive methods as replacements. Firstly, we use an off-the-shelf clustering algorithm, GRACLUS[60], as a baseline (a). Secondly, we replace it with a simple average pooling method used in CDConv[12] as a baseline (b). Lastly, we replace the attention score $\gamma_k^n$ with a random value as a baseline (c). As shown in Table 2, our method significantly outperforms all three baselines. Specifically, it surpasses baseline (a) by an impressive **11.5**%, baseline (b) by **2.5**%, and baseline (c) by **10.2**%. The superior performance compared to baseline (a) highlights the importance of using a learnable clustering approach for effective representation learning. This demonstrates that our neural clustering is able to capture meaningful patterns and structures in the protein that are not captured by the off-the-shelf clustering algorithm. Furthermore, the comparison with baseline (c) supports the notion that learned assignment is more effective than random assignment. This suggests that neural clustering is able to leverage the inherent structure and relationships within the protein to make informed assignments, leading to improved performance.

**Efficiency.** We conduct an investigation into the efficiency of our neural clustering-based framework, focusing on its running time for enzyme reaction classification. The mean running time per prediction was measured using a single Nvidia RTX 3090 GPU and Intel(R) Xeon(R) Gold 6326 CPU @ 2.90GHz, and the results are summarized in Table 3. In particular, GearNet [10], a competing method known for its high complexity, cannot be trained using the same GPU due to its resource requirements (*OOM*). Notably, our method achieves state-of-the-art performance while maintaining a comparable running time to existing approaches, suggesting the efficiency of our method.

**Initial Clustering Radius.** The initial clustering radius $r$ determines the cluster size formed in SCI step. A larger radius leads to more amino acids in each cluster, potentially capturing a greater amount of spatial structural information. However, this also introduces the risk of increased noise within the clusters. Conversely, a smaller radius results in fewer amino acids being included in each cluster, which can reduce noise but may also lead to the loss of some critical information. Therefore, we conducted experiments ranging from 2.0 to 7.0 and assessed the performance on two tasks: protein fold classification and enzyme reaction classification. The experimental results, presented in Fig. 3, indicate that the optimal performance is achieved when $r = 4.0$, which suggests a suitable balance between capturing sufficient structural informa-

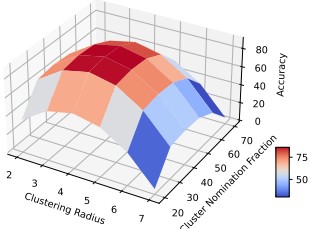

Figure 3: Performance change curve with different combinations of $\omega$ and $r$ for enzyme reaction classification.

tion and mitigating the detrimental effects of noise. More specific results can be found in Table 9.

**Cluster Nomination Fraction.** In CN step, a cluster nomination fraction $\omega$ determines the proportion of clusters selected as the medoid nodes for the next iteration. A larger $\omega$ means that more clusters are retained, which may preserve more information but also increase redundancy and complexity. While a smaller $\omega$ means that fewer clusters are retained, which may reduce redundancy and complexity but also lose some information. We experiment with different values of $\omega$ from 20% to 70% and report results on protein fold classification and enzyme reaction classification. As shown in Fig. 3, the best performance is achieved when $\omega = 40\%$, suggesting a suitable trade-off between preserving information and reducing redundancy and complexity. See Table 9 for more results.

**Number of Iterations.** Table 4 also studies the impact of the number of iterations. For enzyme reaction classification, increasing $T$ from 1 to 4 leads to better performance (*i.e.*, 84.7%→89.6%). However, the accuracy drops significantly from 89.6% to 86.3% when $T$ is set as 5. This may be because over-clustering finds insufficient and insignificant amino acids, which are harmful to representation learning. Similar trends can also be observed in the results of other tasks.

Table 4: Analysis of adopting different number of iterations (§4.5).

| $T$ | EC | GO | | | Fold Classification | | | | Reaction |
|---|---|---|---|---|---|---|---|---|---|
| | | BP | MF | CC | Fold | Super. | Fam. | Avg. | |
| 1 | 0.717 | 0.402 | 0.593 | 0.432 | 55.7 | 73.2 | 97.4 | 75.4 | 84.7 |
| 2 | 0.824 | 0.438 | 0.642 | 0.453 | 60.0 | 79.2 | 99.0 | 79.1 | 88.1 |
| 3 | 0.855 | 0.469 | 0.677 | 0.480 | 62.2 | 80.8 | 99.3 | 80.8 | 89.0 |
| 4 | **0.866** | **0.474** | **0.675** | **0.483** | **63.1** | **81.2** | **99.6** | **81.3** | **89.6** |
| 5 | 0.809 | 0.423 | 0.605 | 0.455 | 58.1 | 75.7 | 98.5 | 77.4 | 86.3 |

Table 5: Different $u\%$ missing coordinates (§4.5).

| $u\%$ | Fold | Super. | Fam. |
|---|---|---|---|
| 0% | **63.1** | **81.2** | **99.6** |
| 5% | 61.9 | 79.8 | 99.5 |
| 10% | 60.1 | 78.7 | 99.5 |
| 20% | 56.7 | 76.9 | 99.3 |
| 30% | 50.2 | 73.6 | 99.2 |
| 40% | 47.8 | 71.3 | 99.0 |

Table 6: Analysis of the impact of rotation invariance and different numbers of CRE blocks (§4.5).

| Rotation Invariant | $B$ | EC | GO | | | Fold Classification | | | | Reaction |
|---|---|---|---|---|---|---|---|---|---|---|
| | | | BP | MF | CC | Fold | Super. | Fam. | Avg. | |
| ✔ | 1 | 0.825 | 0.430 | 0.618 | 0.464 | 57.7 | 76.3 | 99.4 | 77.8 | 87.6 |
| ✔ | 2 | **0.866** | **0.474** | **0.675** | **0.483** | **63.1** | **81.2** | **99.6** | **81.3** | **89.6** |
| ✔ | 3 | 0.857 | 0.466 | 0.669 | 0.474 | 61.8 | 80.2 | 99.5 | 80.5 | 88.9 |
| ✗ | 2 | 0.781 | 0.392 | 0.614 | 0.436 | 56.4 | 75.3 | 97.9 | 76.4 | 87.1 |

**Percentages of Missing Coordinates.** In some cases, the protein structures may have missing coordinates due to experimental errors or incomplete data. To test the robustness of our framework to handle such cases, we randomly remove a certain percentage $u\%$ of coordinates from the protein structures and evaluate our framework on protein fold classification. The results are shown in Table 5, where we can find that our framework still achieves competitive performance when some of the coordinates are missing. For instance, on the Superfamily evaluation scenario, our framework achieves an average accuracy of 78.7% when 10% of the coordinates are missing, which is only slightly lower than the accuracy of 81.2% when no coordinates are missing. This indicates that our framework can learn robust representations of proteins that are not sensitive to missing coordinates.

**Number of CRE Blocks.** In our framework, we use $B$ CRE blocks at each clustering iteration to extract cluster features. We study the impact of using different values of $B$ from 1 to 3 on all four sub-tasks. We stop using $B > 3$ as the required memory exceeds the computational limit of our hardware. The results are shown in Table 6, where we can find that $B = 2$ achieves the best performance on all tasks. For instance, on enzyme reaction classification, $B = 2$ achieves an accuracy of 89.6%, while if $B = 1$ or $B = 3$, the accuracy drops to 87.6% and 88.9%, respectively. This suggests that using two CRE blocks is sufficient to capture the cluster information and adding more blocks does not bring significant improvement but increases the computational cost.

**Rotation Invariance.** We compare our framework with and without rotation invariance on all four tasks. The results are shown in Table 6, where we can see that rotation invariance improves the performance of our framework on all tasks. For example, on protein fold classification, rotation invariance boosts the average accuracy from 76.4% to 81.3%. This indicates that rotation invariance can help our framework to capture the geometric information of proteins more effectively and robustly.

## 4.6 VISUALIZATION

We visualize the protein structure at each iteration in Fig. 4. The color of the node corresponds to the score calculated in CN step. By using such an iterative clustering algorithm, this method is

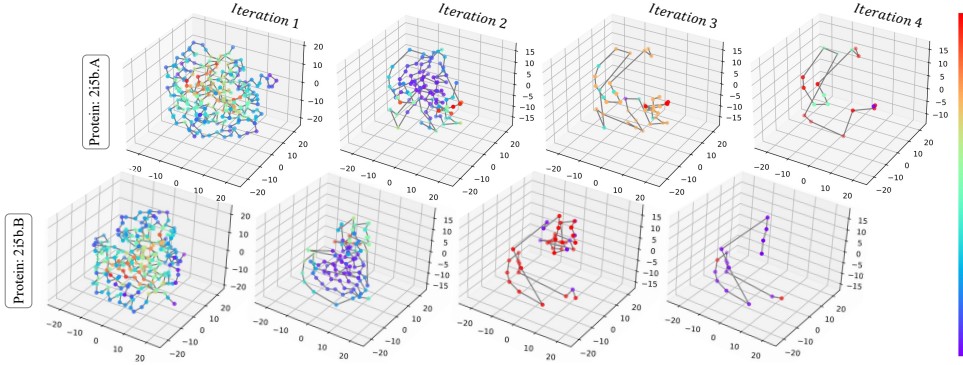

Figure 4: Visualization results of the protein structure at each iteration. The color of the node denotes the score calculated in CN step. See related analysis in §4.6.

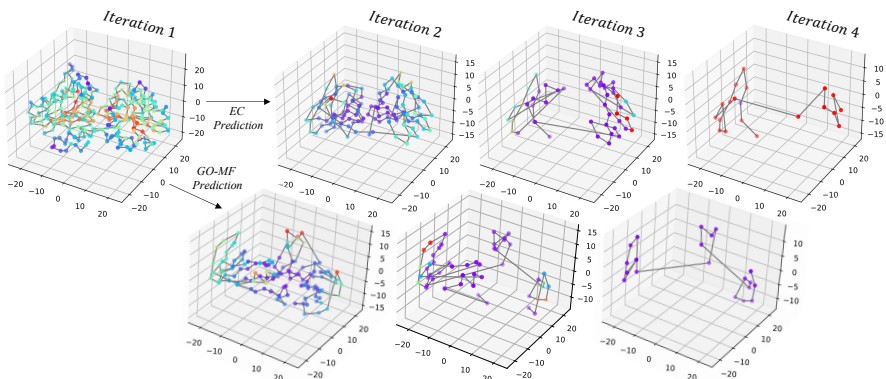

Figure 5: Clustering results for a protein exhibit variations across EC and GO-MF predictions. (§4.6)

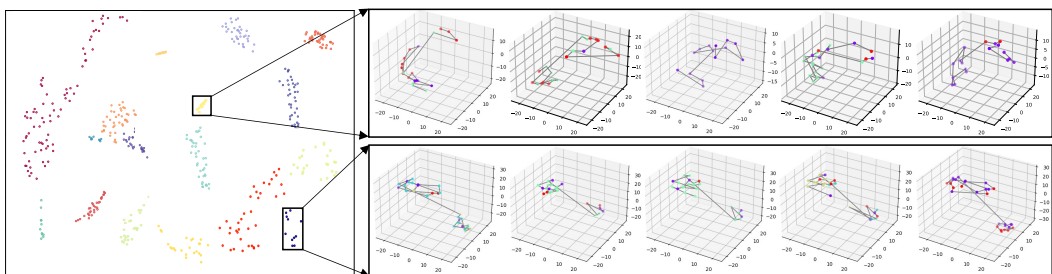

Figure 6: UMAP projection [61] of the learned representation. See related analysis in §4.6.

supposed to explore the critical amino acids of the protein. For example, we examine the protein '2i5b.A', characterized by a complex structure consisting of numerous loops and helices. After the first iteration of clustering, our method selects some amino acids that are located at the ends or bends of these loops and helices as the medoid nodes for the next iteration. Subsequently, after the second clustering iteration, our method further narrows down the number of amino acids by selecting those that have high scores. Ultimately, our method identifies a small subset of amino acids with the highest scores, which are seen as the representative ones for the protein. When comparing the visualization results of protein chain pairs stemming from the same family or protein, *e.g.*, '2i5b.A' *vs* '2i5b.B', we observe remarkable similarity in their clustering outcomes, suggesting that they possess critical amino acids fundamentally determining their respective structures and functionalities. This further validates that our method is effective in identifying these critical amino acids.

In Fig. 5, we present the clustering results of the same protein for different tasks: EC and GO-MF. Interestingly, we observe variations in the results for the same protein across different tasks, indicating that different critical components are identified for different functions. Moreover, some parts are highlighted for both two tasks. This is probably because these parts are informative across tasks. To further prove that our method can indeed discover some functional motifs of proteins, following LM-GV [55], we apply UMAP [61] to analyze the learned representation at the penultimate layer on Enzyme reaction classification and use DBSCAN32 [62] to extract protein clusters. 20 of 384 classes are shown in Fig. 6, where two clusters are selected for detailed analysis. Remarkably, it is clear that proteins originating from the same cluster exhibit similar structural motifs, as generated by our method. This compelling observation underscores the efficacy of our clustering approach in identifying proteins possessing analogous structural motifs that are related to their enzyme reaction functions.

## 5 CONCLUSION

In this work, our epistemology is centered on the protein representation learning by a neural clustering paradigm, which coins a compact and powerful framework to unify the community of protein science and respect the distinctive characteristics of each sub-task. The clustering insight leads us to introduce new approaches for spherosome cluster initialization, cluster representation extraction, and cluster nomination based on both 1D and 3D information of amino acids. Empirical results suggest that our framework achieves superior performance in all four sub-tasks. Our research may potentially benefit the broader domain of bioinformatics and computational biology as a whole.

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

## SUMMARY OF THE APPENDIX

This appendix contains additional details for the ICLR 2024 submission, titled *"Clustering for Protein Representation Learning"*. The appendix provides more details of our approach, additional literature review, further discussions, additional experimental results, information on broader impacts, and limitations. These topics are organized as follows:

- §A: Details of Training Setup
- §B: Details of Evaluation Metrics
- §C: Clustering Algorithm
- §D: Rotation Invariance
- §E: Additional Literature Review
- §F: More Quantitative Results
- §G: More Qualitative Results
- §H: Broader Impacts
- §I: Limitations

## A    DETAILS OF TRAINING SETUP

In our experiments, we train our framework using a SGD optimizer with learning rate of 1e-3 and a weight decay of 5e-4. Due to memory constraints, we set the batch size to 32, 24, 8, and 8 for EC, GO, Fold Classification, and Reaction tasks, respectively. The framework comprises four iterations of clustering, which employ varying numbers of channels at each iteration. For Fold Classification, we use 256, 512, 1024 and 2048 channels for the four iterations, respectively. For EC, GO and Reaction, we use 128, 256, 512 and 1024 channels for the four iterations, respectively. The training process spans 200 or 300 epochs for each dataset, and the best model is selected based on the validation performance. Details can be seen in Table 7.

In addition, we adopt data augmentation techniques, similar to those used in [9, 12] to augment the data for fold and reaction classification tasks. Specifically, we apply Gaussian noise with a standard deviation of 0.1 and anisotropic scaling within the range of [0.9, 1.1] to the amino acid coordinates in the input data. We also add the same noise to the atomic coordinates within the same amino acid to ensure that the internal structure of each amino acid remains unchanged.

Table 7: The hyperparameter configurations of our method vary across different tasks. We choose all the hyperparameters based on their performance on the validation set. See details in §A.

| Hyperparameter | EC | GO-BP | GO-MF | GO-CC | Fold Classification | Reaction |
|---|---|---|---|---|---|---|
| batch size | 32 | 24 | 24 | 24 | 8 | 8 |
| Channels | [128,256,512,1024] | [128,256,512,1024] | [128,256,512,1024] | [128,256,512,1024] | [256,512,1024,2048] | [128,256,512,1024] |
| # epoch | 300 | 300 | 300 | 300 | 200 | 200 |

## B    DETAILS OF EVALUATION METRICS

We first present the details of evaluation metrics for enzyme commission number prediction and gene ontology term prediction. The objective of these tasks is to determine whether a protein possesses specific functions, which can be viewed as multiple binary classification tasks. We define the first metric as the protein-centric maximum $F$-score ($F_{\max}$). This score is obtained by calculating the precision and recall for each protein and then averaging the scores over all proteins. To be more specific, for a given target protein $i$ and a decision threshold $\lambda \in [0, 1]$, we compute the precision and recall as follows:

$$\text{precision}_i(\lambda) = \frac{\sum_a \mathbb{1}[a \in P_i(\lambda) \cap G_i]}{\sum_a \mathbb{1}[a \in P_i(\lambda)]}, \quad \text{recall}_i(\lambda) = \frac{\sum_a \mathbb{1}[a \in P_i(\lambda) \cap G_i]}{\sum_a \mathbb{1}[a \in G_i]},$$

where $a$ represents a function term in the ontology, $G_i$ is a set of experimentally determined function terms for protein $i$, $P_i(\lambda)$ denotes the set of predicted terms for protein $i$ with scores greater than or equal to $\lambda$, and $\mathbb{1}[\cdot] \in \{0, 1\}$ is an indicator function that is equal to 1 if the condition is true.

Then, the average precision and recall over all proteins at threshold $\lambda$ is defined as:

$$\text{precision}(\lambda) = \frac{\sum_i \text{precision}_i(\lambda)}{M(\lambda)}, \quad \text{recall}(\lambda) = \frac{\sum_i \text{recall}_i(\lambda)}{N},$$

where we use $N$ to represent the number of proteins, and $M(\lambda)$ to denote the number of proteins on which at least one prediction was made above threshold $\lambda$, $i.e.$, $|P_i(\lambda)| > 0$. By combining these two measures, the maximum F-score is defined as the maximum $F$-measure value obtained across all thresholds:

$$F_{\max} = \max_{\lambda} \left\{ \frac{2 \times \text{precision}(\lambda) \times \text{recall}(\lambda)}{\text{precision}(\lambda) + \text{recall}(\lambda)} \right\}.$$

The second metric, mean accuracy, is calculated as the average precision scores for all protein-function pairs, which is equivalent to the micro average precision score for multiple binary classification.

## C    CLUSTERING ALGORITHM

---

**Algorithm 1:** Pseudo-code of our neural clustering algorithm

**Input**          : Protein $\mathcal{P} = (\mathcal{V}, \mathcal{E}, \mathcal{Y})$; Amino acid embedding $\boldsymbol{e}_j$ for amino acid $v_j \in \mathcal{V}$; Cluster nomination ratio $\omega$; Nomination operator NOMINATE; Index selection operator INDEXSELECT; Add self-loop operator ADDSELFLOOPS; Spherosome clustering operator RADIUS; Spherosome clustering radius $r$; ReLU activation function $\sigma$; Geometric coordinates $Pos$; Geometric orientations $Ori$; Sequential orders $Seq$

**Intermediate:**  Clustered features $X^c$ and Scored cluster features $\hat{X}^c$; Adjacency matrix $A$; Edge index $E$; Cluster scores vector $\Phi$; Nominated index $index$

**Output**         : Nominated amino acid features $X$, coordinates $Pos$, orientations $Ori$, sequential orders $Seq$

1 **for** $t = 1, 2, 3, 4$ **do**
2    $A \leftarrow \text{RADIUS}(Pos, r)$;
3    $E \leftarrow \text{ADDSELFLOOPS}(A)$;
4    **for** $n = 1...N_{t-1}$ **do**
5       $\tilde{\boldsymbol{x}}_n \leftarrow \vec{0}$;
6       **for** $k = 1...K$ **do**
7          $\boldsymbol{g}_k^n, \boldsymbol{o}_k^n, d_k^n, s_k \leftarrow (Pos, Ori, Seq)$;
8          $\boldsymbol{x}_k^n \leftarrow f(\boldsymbol{g}_k^n, \boldsymbol{o}_k^n, d_k^n, s_k, \boldsymbol{e}_k)$;
9          $\gamma_k^n \leftarrow softmax(\sigma([\boldsymbol{W}_1 \boldsymbol{x}_n, \boldsymbol{x}_k^n]))$;
10         $\tilde{\boldsymbol{x}}_n \leftarrow \tilde{\boldsymbol{x}}_n + \gamma_k^n \boldsymbol{x}_k^n$;
11       **end**
12       $X_n^c \leftarrow \tilde{\boldsymbol{x}}_n$;
13    **end**
14    $\Phi \leftarrow \text{GCN}(X^c, E)$;
15    $\hat{X}^c \leftarrow \Phi \odot X^c$;
16    $N_t \leftarrow \lfloor \omega \cdot N_{t-1} \rfloor$;
17    $index \leftarrow \text{NOMINATE}(\Phi, N_t)$;
18    $X, Pos, Ori, Seq \leftarrow \text{INDEXSELECT}(\hat{X}^c, Pos, Ori, Seq, index)$;
19 **end**

---

## D    ROTATION INVARIANCE

**Rotation Invariance**. To make our method rotationally invariant, we augment the distance information $d_k^n$ by using a relative spatial encoding [53]:

$$d_k^n = \left( d(||\boldsymbol{z}_k - \boldsymbol{z}_n||), \ \boldsymbol{O}_n^\top \frac{\boldsymbol{z}_k - \boldsymbol{z}_n}{||\boldsymbol{z}_k - \boldsymbol{z}_n||}, \ q(\boldsymbol{O}_n^\top \boldsymbol{O}_k^n) \right), \tag{6}$$

where $\boldsymbol{O}_n = [\boldsymbol{b}_n, \boldsymbol{j}_n, \boldsymbol{b}_n \times \boldsymbol{j}_n]$, $\boldsymbol{b}_n$ denotes the negative bisector of angle between the ray $(v_{n-1} - v_n)$ and $(v_{n+1} - v_n)$, and $\boldsymbol{j}_n$ is a unit vector normal to that plane. Formally, we have $\boldsymbol{u}_n = \frac{\boldsymbol{z}_n - \boldsymbol{z}_{n-1}}{||\boldsymbol{z}_n - \boldsymbol{z}_{n-1}||} \in \mathbb{R}^3$, $\boldsymbol{b}_n = \frac{\boldsymbol{u}_n - \boldsymbol{u}_{n+1}}{||\boldsymbol{u}_n - \boldsymbol{u}_{n+1}||} \in \mathbb{R}^3$, $\boldsymbol{j}_n = \frac{\boldsymbol{u}_n \times \boldsymbol{u}_{n+1}}{||\boldsymbol{u}_n \times \boldsymbol{u}_{n+1}||} \in \mathbb{R}^3$, where $\times$ is the cross product. The first term in Eq. 6 is a distance encoding $d(\cdot)$ lifted into the radius $r$, the second term is a direction encoding that corresponds to the relative direction of $v_k^n \to v_n$, and the third term is an orientation encoding $q(\cdot)$ of the quaternion representation of the spatial rotation matrix. This encoding approach allows us to capture both local and global geometric information while being invariant to different orientations. Related experimental results are seen in Table 6.

## E    ADDITIONAL LITERATURE REVIEW

**Graph Pooling.** Graph pooling designs have been proposed to achieve a useful and rational graph representation. These designs can be broadly categorized into two types [63]: Flat Pooling [64–68] and Hierarchical Pooling [69–72]. Flat Pooling generates a graph-level representation in a single step by primarily calculating the average or sum of all node embeddings without consideration of the intrinsic hierarchical structures of graphs, which causes information loss [73]. On the other hand, Hierarchical Pooling gradually reduces the size of the graph. Previous graph pooling algorithms, as variants of GCN, still follow the message passing pipeline. Typically, they are hard to be trained and need many extra regularizations and/or operations. For example, Diffpool [69] is trained with an auxiliary link prediction objective. Besides, it generates a dense assignment matrix thus incurring a quadratic storage complexity. Top-K pooling [70], though also addressing the selection of top nodes, adopts a Unet-like, graph encoder-decoder architecture, which is much more complicated than our model but only learns a simple scalar projection score for each node.

In contrast, our clustering-based algorithm is more principled and elegant. It can address the sparsity concerns of Diffpool and capture rich protein structure information by aggregating amino acids to form clusters instead of learning from a single node. Furthermore, it sticks to the principle of clustering throughout its algorithmic design: SCI step is to form the clusters by considering geometrical relations among amino acids; CRE step aims to extract cluster-level representations; CN step is for the selection of important cluster centers. It essentially combines unsupervised clustering with supervised classification. The forward process of our model is inherently a neural clustering process, which is more transparent and without any extra supervision.

## F    MORE QUANTITATIVE RESULTS

Table 8: $F_{\max}$ on GO term and EC number prediction under different cutoffs. See details in §F.

| Cutoff | 30% | 40% | 50% | 70% | 95% | 30% | 40% | 50% | 70% | 95% |
|---|---|---|---|---|---|---|---|---|---|---|
| Method | GO-BP | | | | | GO-MF | | | | |
| CNN [74] | 0.197 | 0.195 | 0.197 | 0.211 | 0.244 | 0.238 | 0.243 | 0.256 | 0.292 | 0.354 |
| ResNet [57] | 0.230 | 0.230 | 0.234 | 0.249 | 0.280 | 0.282 | 0.288 | 0.308 | 0.347 | 0.405 |
| LSTM [57] | 0.194 | 0.192 | 0.195 | 0.205 | 0.225 | 0.223 | 0.229 | 0.245 | 0.276 | 0.321 |
| Transformer [57] | 0.267 | 0.265 | 0.262 | 0.262 | 0.264 | 0.184 | 0.187 | 0.195 | 0.204 | 0.211 |
| GCN [58] | 0.251 | 0.250 | 0.248 | 0.248 | 0.252 | 0.180 | 0.183 | 0.187 | 0.194 | 0.195 |
| GearNet [10] | 0.345 | 0.347 | 0.354 | 0.378 | 0.403 | 0.444 | 0.461 | 0.490 | 0.537 | 0.580 |
| CDConv [12] | 0.381 | 0.390 | 0.401 | 0.428 | 0.453 | 0.533 | 0.553 | 0.577 | 0.621 | 0.654 |
| Ours | **0.390** | **0.397** | **0.405** | **0.429** | **0.474** | **0.551** | **0.569** | **0.595** | **0.629** | **0.675** |
| Method | GO-CC | | | | | EC | | | | |
| CNN [74] | 0.258 | 0.257 | 0.260 | 0.263 | 0.387 | 0.366 | 0.361 | 0.372 | 0.429 | 0.545 |
| ResNet [57] | 0.277 | 0.273 | 0.280 | 0.278 | 0.304 | 0.409 | 0.412 | 0.450 | 0.526 | 0.605 |
| LSTM [57] | 0.263 | 0.264 | 0.269 | 0.270 | 0.283 | 0.247 | 0.249 | 0.270 | 0.333 | 0.425 |
| Transformer [57] | 0.378 | 0.382 | 0.388 | 0.395 | 0.405 | 0.167 | 0.173 | 0.175 | 0.197 | 0.238 |
| GCN [58] | 0.318 | 0.318 | 0.320 | 0.323 | 0.329 | 0.245 | 0.246 | 0.246 | 0.280 | 0.320 |
| GearNet [10] | 0.394 | 0.394 | 0.401 | 0.408 | 0.450 | 0.625 | 0.646 | 0.694 | 0.757 | 0.810 |
| CDConv [12] | 0.428 | 0.435 | 0.440 | 0.451 | 0.479 | 0.634 | 0.659 | 0.702 | 0.768 | 0.820 |
| Ours | **0.433** | **0.442** | **0.449** | **0.457** | **0.483** | **0.713** | **0.759** | **0.795** | **0.837** | **0.866** |

**Different Sequence Cutoffs.** Experiments in Table 1 use 95% as the sequence identity cutoff for EC and GO dataset splitting. In addition, we test our method under four lower sequence identity cutoffs (30%/40%/50%/70%) following [8] and demonstrate the experimental results in Table 8. The purpose of employing various sequence identity cutoffs is to assess the robustness of different models

Table 9: Analysis of initial clustering radius and nomination fraction. See related analysis in §4.5

| Radius $r$ | Fold | Super. | Fam. | Reaction | Fraction $\omega$ | Fold | Super. | Fam. | Reaction |
|---|---|---|---|---|---|---|---|---|---|
| 2.0 | 51.2 | 67.3 | 96.5 | 85.2 | 20% | 53.2 | 70.5 | 98.3 | 85.9 |
| 3.0 | 60.3 | 78.9 | 99.5 | 89.1 | 30% | 62.7 | 81.1 | 99.6 | 89.4 |
| 4.0 | **63.1** | **81.2** | 99.6 | **89.6** | 40% | **63.1** | **81.2** | 99.6 | **89.6** |
| 5.0 | 62.2 | 80.7 | 99.6 | 89.1 | 50% | 61.0 | 80.6 | 99.6 | 88.9 |
| 6.0 | 59.4 | 79.9 | 99.4 | 88.2 | 60% | 61.3 | 79.9 | 99.4 | 88.5 |
| 7.0 | 56.9 | 71.2 | 99.3 | 87.7 | 70% | 60.8 | 78.6 | 99.3 | 88.3 |

Table 10: Comparison results with existing protein language models. See details in §F.

| Method | Pretraining Dataset | | EC | GO | | | Fold Classification | | | | Reaction |
|---|---|---|---|---|---|---|---|---|---|---|---|
| | | | | BP | MF | CC | Fold | Super. | Fam. | Avg. | |
| DeepFRI [75] | Pfam | 10M | 0.631 | 0.399 | 0.465 | 0.460 | 15.3 | 20.6 | 73.2 | 36.4 | 63.3 |
| ESM-1b [76] | UniRef50 | 24M | 0.864 | 0.470 | 0.657 | 0.488 | 26.8 | 60.1 | 97.8 | 61.6 | 83.1 |
| ProtBERT-BFD [77] | BFD | 2.1B | 0.838 | 0.279 | 0.456 | 0.408 | 26.6 | 55.8 | 97.6 | 60.0 | 72.2 |
| IEConv (amino level) [54] | PDB | 476K | - | 0.468 | 0.661 | 0.516 | 50.3 | 80.6 | 99.7 | 76.9 | 88.1 |
| LM-GVP [55] | UniRef100 | 0.21B | 0.664 | 0.417 | 0.545 | 0.527 | - | - | - | - | - |
| GearNet-Edge-IEConv [10] | AlphaFoldDB | 805K | 0.874 | 0.490 | 0.654 | 0.488 | 54.1 | 80.5 | 99.9 | 78.2 | 87.5 |
| IEConv (residue level) [54] | - | | - | 0.421 | 0.624 | 0.431 | 47.6 | 70.2 | 99.2 | 72.3 | 87.2 |
| GearNet-Edge-IEConv [10] | - | | 0.810 | 0.403 | 0.580 | 0.450 | 48.3 | 70.3 | 99.5 | 72.7 | 86.6 |
| CDConv [12] | - | | 0.820 | 0.453 | 0.654 | 0.479 | 56.7 | 77.7 | 99.6 | 78.0 | 88.5 |
| Ours | - | | 0.866 | 0.474 | **0.675** | 0.483 | **63.1** | **81.2** | 99.6 | **81.3** | **89.6** |

when subjected to diverse hold-out test sets. Lowering the cutoff value indicates a lower degree of similarity between the training and test sets. Notably, we observe that even at lower cutoff values, our model consistently achieves the highest performance. This finding underscores the robustness and generalization capability of our model, as it demonstrates superior performance even when confronted with test sets that exhibit lower similarity to the training data. For example, in EC number prediction under cutoff 30%, our method outperforms CDConv [12] by 12.5% (0.634→0.713).

**Comparison to existing protein language models.** Existing protein language models [10, 54, 55, 76, 77] are typically pre-trained with much more data. The network design, training objective, training data, as well as training protocol are greatly different. Our method aims to learn protein representation through a clustering scheme that combines unsupervised clustering with supervised classification. Our contributions are vertical. Thus making a direct comparison seems a little bit unfair. To further showcase the effectiveness of our method, we compare our algorithm with some recent protein pretraining language models on fold classification. As shown in Table 10, our method still yields better results without any pre-training or self-supervised learning. It may be that one or a few self-supervised tasks are insufficient to learn effective representations, as mentioned in [12]. This also sheds light on the direction of our future efforts: it is interesting to incorporate our algorithm with existing protein language models, as our core idea is principled.

**More Datasets and Downstream Tasks.** We conducted additional experiments and discussions on one more newer and comprehensive benchmark (*i.e.*, PROBE [78]) and three additional downstream tasks to demonstrate the effectiveness of our method. These tasks include protein inverse folding, protein-ligand binding affinity prediction, and protein-protein interactions. The results of these experiments are presented in Table 11, Table 12, Table 13, and Table 14. It is worth noting that our method consistently outperforms existing state-of-the-art approaches on all datasets and tasks. For instance, in the inverse folding task, our method achieves a 3.19% improvement in Recovery-All compared to the current state-of-the-art method, PiFold [79]. Similarly, in the protein-ligand binding affinity prediction task, our method outperforms the existing state-of-the-art method, Atom3D [29], by 5.26% (0.553→0.582) on the sequence identity 30% split. Additionally, in the protein-protein interactions task, our method achieves a 2.07% improvement compared to ProNet [11]. These results highlight the superior performance of our method across multiple downstream tasks. The consistent improvements obtained provide solid evidence for the effectiveness of our overall algorithm design.

# G  MORE QUALITATIVE RESULTS

We provide more visualization results of the clustering results at each iteration in Fig. 7. The color of the node denotes the score calculated in our CN step. For example, in the first row, we can see that the protein '1rco.E' has a helical structure with some loops. After the first iteration of clustering, our method selects some amino acids that are located at the ends or bends of these loops

Table 11: Results on the protein inverse folding task. (§F)

| Method | Perplexity ↓ | | | Recovery % ↑ | | |
|---|---|---|---|---|---|---|
| | Short | Single-chain | All | Short | Single-chain | All |
| GCA [80] | 7.09 | 7.49 | 6.05 | 32.62 | 31.10 | 37.64 |
| GVP [81] | 7.23 | 7.84 | 5.36 | 30.60 | 28.95 | 39.47 |
| ProteinMPNN [82] | 6.21 | 6.68 | 4.61 | 36.35 | 34.43 | 45.96 |
| PiFold [79] | 6.04 | 6.31 | 4.55 | 39.84 | 38.53 | 51.66 |
| Ours | **5.89** | **5.77** | **4.37** | **44.47** | **44.29** | **54.85** |

Table 12: Results on the protein-protein interactions task. (§F)

| Method | AUROC ↑ |
|---|---|
| Atom3D [29] | 0.844 |
| GVP-GNN [33] | 0.866 |
| ProNet [11] | 0.871 |
| Ours | **0.889** |

Table 13: Results on protein-ligand binding affinity prediction task. See details in §F.

| Method | Sequence Identity 30% | | | Sequence Identity 60% | | |
|---|---|---|---|---|---|---|
| | RMSE ↓ | Pearson ↑ | Spearman ↑ | RMSE ↓ | Pearson ↑ | Spearman ↑ |
| Atom3D [29] | **1.416** | 0.550 | 0.553 | 1.621 | 0.608 | 0.615 |
| DeepDTA [75] | 1.866 | 0.472 | 0.471 | 1.762 | 0.666 | 0.663 |
| TAPE [57] | 1.890 | 0.338 | 0.286 | 1.633 | 0.568 | 0.571 |
| ProtTrans [77] | 1.544 | 0.438 | 0.434 | 1.641 | 0.595 | 0.588 |
| MaSIF [83] | 1.484 | 0.467 | 0.455 | 1.426 | 0.709 | 0.701 |
| IEConv [9] | 1.554 | 0.414 | 0.428 | 1.473 | 0.667 | 0.675 |
| ProNet [11] | 1.463 | 0.551 | 0.551 | 1.343 | 0.765 | 0.761 |
| Ours | 1.427 | **0.578** | **0.582** | **1.339** | **0.773** | **0.766** |

and helices as the center nodes for the next iteration. These amino acids may play an important role in stabilizing the protein structure or interacting with other molecules. After the second iteration of clustering, our method further narrows down the number of center nodes by selecting those that have high scores. These amino acids may form functional domains or motifs that are essential for the protein function. Our method finally identifies a few amino acids that have the highest scores and are most representative of the protein structure and function. Through visualizing the clustering results at each iteration, we can explicitly understand how our method progressively discovers the critical components of different proteins by capturing their structural features in a hierarchical way.

In addition, as shown in the figure, we present pairs of protein chains from the same family or same protein (*i.e.*, '1rco.R' and '1rco.E', '3n3y.B' and '3n3y.C', '6gk9.B' and '6gk9.D'). For example, we observe that the clustering results of '3n3y.B' and '3n3y.C' (two chains of the same protein) are very similar, indicating that they have similar critical amino acids that determine their structure and function. This observation is consistent with the biological reality that proteins from the same family or same protein often have long stretches of similar amino acid sequences within their primary structure, suggesting that our method is effective in identifying these critical amino acids.

## H  BROADER IMPACTS

Our neural clustering framework for protein representation learning has several potential applications and implications for society. Protein representation learning can help advance our understanding of protein structure and function, which are essential for many biological processes and diseases. By discovering the critical components of proteins, our framework can facilitate protein design, engineering, and modification, which can lead to the development of novel therapies, drugs, and biotechnologies. For example, our framework can assist in designing new protein sequences that exhibit specific properties or functions, such as catalyzing biochemical reactions or binding to other molecules. This can enable the creation of new enzymes, antibodies, vaccines, and biosensors that can have a positive impact on human health and well-being.

However, our framework also poses some ethical and social challenges that need to be addressed. Protein representation learning may raise issues of data privacy, ownership, and security, as protein data may contain sensitive information about individuals or organisms. For example, protein data may reveal genetic information, disease susceptibility, or drug response of a person or a population. Therefore, proper measures need to be taken to protect the privacy and security of protein data and prevent unauthorized access or misuse. Moreover, protein representation learning may have unintended consequences or risks for the environment and society, as protein design and engineering may create novel or modified proteins that have unknown or harmful effects. For instance, protein design and engineering may introduce new allergens, toxins, or pathogens that can affect human

Table 14: Results on an additional benchmark (PROBE [78]) for protein representation learning. (§F)

| Method | Pretraining Dataset | | Semantic similarity inference | | | | Ontology-based PFP | | | | Protein family classification | | | |
|---|---|---|---|---|---|---|---|---|---|---|---|---|---|---|
| | | | MF | BP | CC | Avg. | MF | BP | CC | Avg. | Random | 50% | 30% | 15% |
| ProtVec [22] | - | | 0.19 | 0.30 | 0.21 | 0.23 | 0.64 | 0.36 | 0.38 | 0.46 | 0.34 | 0.31 | 0.39 | 0.37 |
| Learned-Vec [23] | - | | 0.41 | 0.30 | 0.31 | 0.34 | 0.68 | 0.39 | 0.41 | 0.49 | 0.59 | 0.60 | 0.58 | 0.54 |
| CPCProt [84] | - | | 0.06 | 0.11 | -0.09 | 0.03 | 0.65 | 0.40 | 0.44 | 0.50 | 0.63 | 0.66 | 0.62 | 0.64 |
| MSA-Transformer [85] | UniRef50 | 24M | 0.38 | 0.31 | 0.30 | 0.33 | 0.67 | 0.47 | 0.50 | 0.55 | 0.67 | 0.72 | 0.73 | 0.63 |
| ProtBERT-BFD [86] | BFD | 2.1B | 0.29 | 0.32 | 0.42 | 0.34 | 0.85 | 0.61 | 0.62 | 0.69 | 0.84 | 0.84 | 0.84 | 0.81 |
| ESM-1b [76] | UniRef50 | 24M | 0.38 | 0.42 | 0.37 | 0.39 | 0.83 | 0.53 | 0.61 | 0.66 | 0.87 | 0.84 | 0.92 | 0.86 |
| ProtXLNet [86] | UniRef100 | 0.21B | 0.23 | 0.31 | 0.25 | 0.26 | 0.82 | 0.50 | 0.59 | 0.63 | 0.81 | 0.80 | 0.85 | 0.72 |
| ProtALBERT [86] | UniRef100 | 0.21B | 0.22 | 0.37 | 0.32 | 0.30 | 0.89 | 0.63 | 0.64 | 0.72 | 0.92 | 0.91 | 0.92 | 0.88 |
| ProtT5-XL [86] | BFD | 2.1B | **0.57** | 0.21 | 0.40 | 0.39 | 0.90 | 0.66 | 0.68 | 0.75 | 0.92 | 0.92 | 0.92 | 0.90 |
| Ours | - | | 0.54 | **0.49** | **0.45** | **0.50** | **0.91** | **0.69** | **0.72** | **0.80** | **0.93** | 0.92 | **0.93** | **0.92** |

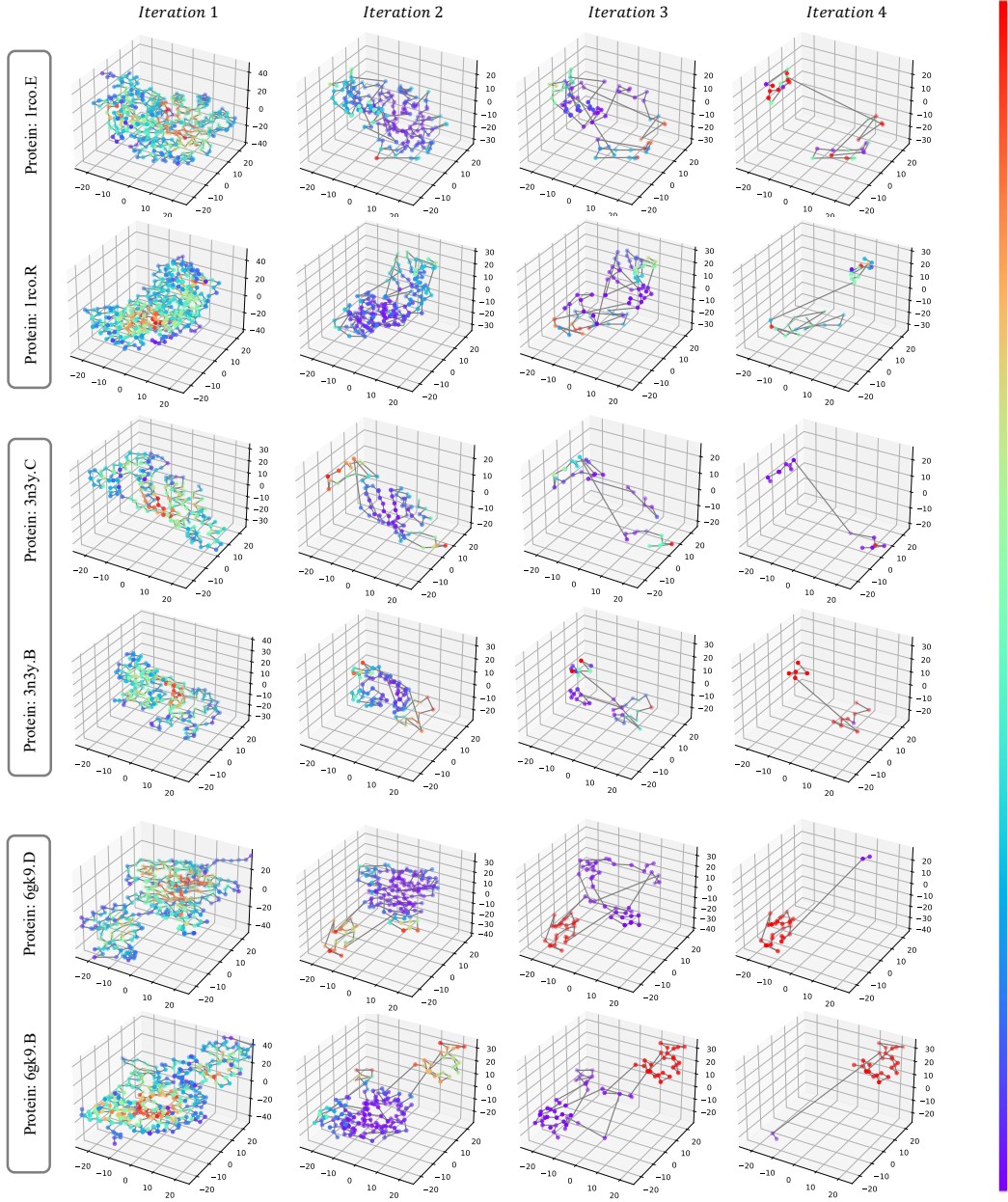

Figure 7: More visualization results. See related analysis in §G.

health or biodiversity. Therefore, proper regulations and oversight need to be established to ensure the safety and responsibility of protein design and engineering.

# I  LIMITATIONS

Our neural clustering framework for protein representation learning has several limitations that need to be addressed in future work. First, our framework relies on the availability of protein structures, which are not always easy to obtain or predict. Although our framework can leverage both sequence-based and structure-based features, it may lose some information that is only encoded in the 3D structure. Second, our framework assumes that the critical components of a protein are determined by its amino acid sequence and structure, but it does not consider other factors that may affect protein function, such as post-translational modifications, interactions with other molecules, or environmental conditions. Third, our framework does not explicitly account for the evolutionary relationships among proteins, which may provide useful information for protein representation learning. Incorporating phylogenetic information into our framework may enhance its ability to capture the functional diversity and similarity of proteins.

