# OpenReview forum: "Clustering for Protein Representation Learning"
_ICLR.cc/2024/Conference — ICLR 2024 Conference Withdrawn Submission_

### Official Review · Reviewer_36dv · 2023-10-26

**Soundness:** 2 fair
**Presentation:** 4 excellent
**Contribution:** 2 fair
**Rating:** 1
**Confidence:** 4

**Summary:**

This paper develops a method for training a supervised learning algorithm to carry out various protein classification tasks, including EC prediction, GO term prediction, and protein fold classification, among others.  The main idea is to use a hierarchical representation of each protein, wherein spherical clusters are iteratively joined to identify key amino acids.  These are then fed to a multilayer perceptron for classification.  The approach is benchmarked on a several standard benchmarks and shows remarkably good performance relative to state-of-the-art methods.  The paper also includes some visualizations of the learned key amino acids.

**Strengths:**

The paper is clearly written and does a good job of explaining the importance of the task.

Many different benchmark datasets are analyzed, and many competing methods are included.

**Weaknesses:**

I think this paper misrepresents what it's about.  The paper claims to be about representation learning, but it's really about protein classification.  The fact that you focus on one particular layer of their network (the one before the MLP) does not actually mean that this layer is any more fundamental than other layers.  To my mind, representation learning should be about learning a representation in an unsupervised fashion that can be used to solve multiple downstream tasks.  Instead, in this method, a new representation is learned for every supervised task.  Indeed, I found it quite surprising, in Section 3.1, that representation learning was framed as a classification task.  This just seems wrong to me.

The introduction also does a poor job of preparing us for what the method actually consists of.  This is fundamentally a deep neural network method, but the intro talks about it as if it's doing multiple found of clustering.

The network itself is under-described.  I don't think anyone reading this paper could even approximately reproduce this work.  An example is that the text mentions self-loops but doesn't mention where they are.

I recognize that you are re-using public benchmarks, but I still find it surprising that the train/test splits allow such high sequence similarity.  In the EC benchmark, for example, a train and test example can be up to 95% identical.  The other benchmarks don't even discuss how train/test leakage is avoided.

It's distressing that you claim that protein fold recognition was introduced by a paper in 2018.  This problem has been studied very extensively for at least 20 years prior to that.  See, e.g., the Jaakkola-Haussler paper that won the best paper award at ISMB in 1999.

This observation leads me to my primary concern about this paper, which is that the results are simply too good to be true.  If you really did improve across the board on all of these tasks, achieving such consistent and significant improvements, then this paper should be published in Nature rather than ICLR.   To justify such remarkable results requires much more extensive validation and explanation, rather than just a single table with summary statistics.  Taking, for example, the fold recognition problem, it would be necessary to examine in detail the predictions made by your method and the best-performing second best model.  I'd like to know which proteins you got right that the other method failed at, and I'd like to understand how your method was able to achieve this gain.  It's also not at all clear to me that the methods in Table 1 really do represent the state of the art.  Some obvious missing contenders are AlphaFold and RosettaFold.


Minor comments:

The abstract talks about "performance" but fails to make clear what problem is being solved.

In the penultimate paragraph of the intro, I think you should elaborate a bit more on what the representation actually consists of.  In particular, it's not clear whether the final MLP receives just the identities of the selected amino acids, or if it also retains some notion of the hierarchy.

The term "spherosome" doesn't seem to be appropriate here.  I had never heard the term, but a web search suggests that it's the name of an organelle.  Just say "spherical."

**Questions:**

Are the sequential edges in your initial implementation actually important?  As long as the sphere is larger than the connection between two adjacent amino acids, then you could just use a sphere.

I was surprised that the input layer of the model contains, among other features, the spatial coordinates of the amino acid.  This seems counterintuitive, because the protein's representation should be invariant to scale, rotation and translation.  Why does this even make sense?

You note that the fact that different proteins from the same family end up selecting the same amino acids as "representative" is only surprising if the proteins are quite diverged evolutionarily.  How diverged is this family, and how well does this analysis when comparing proteins that share function but very little residual sequence identity?

Figure 5 shows that the selected residues differ by task.  Is there any evidence that these residues are actually relevant to their respective tasks?  Just selecting different residues is not, by itself, super surprising.

Figure 6 does not show that the proteins exhibit similar structural motifs.  It simply shows that proteins from the same class have similar representations internally.  Again, is there any evidence that this is not just a reflection of overall sequence similarity?

---

### Official Review · Reviewer_9Yjf · 2023-10-27

**Soundness:** 2 fair
**Presentation:** 1 poor
**Contribution:** 2 fair
**Rating:** 3
**Confidence:** 4

**Summary:**

The paper presents an approach for obtaining representations of proteins through clustering of their amino acids. More specifically, a graph is defined for each protein whose nodes correspond to amino acids and binary edges indicate the sequential or spatial connection between amino acids. An iterative three-step clustering procedure is applied to this graph and the amino acids corresponding to cluster medoids constitute the reduced protein representation at each step.  The effectivess of the obtained representations is tested on several protein related classification and prediction tasks.

**Strengths:**

- The proposed idea based on amino acid clustering seems to constitute an alternative approach for obtaining protein representations.
- The method seems to provide superior experimental results.
- The amino acid based representation gives the opportunity for visualization and explainability.

**Weaknesses:**

- This is an application paper without considerable novelty from a machine learning point of view.
- A major issue with the paper concerns presentation and clarity in section 3 and the Algorithm 1 in Appendix (see the questions section below) that makes difficult to assess the technical integrity of the proposed method.
- The paper does not provide sufficient information (neither in main text nor in the appendix ('graph pooling' section)) to qualitatively compare the proposed approach with existing ones.
- The method contains several hyperparameters that influence performance.

**Questions:**

1) Since a radius value r is used, the number of neighbors of node n is not fixed (denote as K), but depends on n, ie., the notation K_n should be used.
2) It is not clear how e_n is obtained for each amino acid n.
3) Eq. (2) seems rather strange. One should expect to compute the dot product between x_n and x_k^n.
4) A major question is how the weights W involved in eq. (4) are computed.
5) The formula in eq. (4) seems adhoc. The rationale behind this formula should be provided.
6) How do you specify GCN in line 14 of Algorithm 1?
7) It should be clarified if the proposed method for selecting the amino acids (Algorithm 1) is supervised or unsupervised.

---

### Official Review · Reviewer_KtEJ · 2023-10-31

**Soundness:** 3 good
**Presentation:** 3 good
**Contribution:** 4 excellent
**Rating:** 8
**Confidence:** 2

**Summary:**

The paper presents a neural clustering framework for protein representation learning, which aims to automatically identify the critical components of a protein from its amino acid sequences. Each protein is treated as a graph, where each node in this graph corresponds to an amino acid. Using an iterative clustering strategy, the framework groups nodes based on their 1D and 3D positions and assigns scores to each cluster. Subsequent iterations prioritize high-scoring clusters and use their central nodes (medoid nodes) to refine the clustering. The efficacy of this method is demonstrated across four protein-related tasks, showcasing its state-of-the-art performance.

**Strengths:**

- The paper's approach to protein representation using a neural clustering framework is innovative.

- The proposed method achieves state-of-the-art performance on four protein-related tasks, indicating its effectiveness.

- Comprehensive experiments are performed in this study.

**Weaknesses:**

- The stability of visualization is not discussed.

**Questions:**

1. In Section 4.6, the author mentioned “we observe remarkable similarity in their clustering outcomes”, which I think may be subjective. Is there any metric for this similarity measurement?

2. How stable/robust is the visualization? Does the method always present similarity visualization results when we run it multiple times?

---

### Official Review · Reviewer_5mqs · 2023-11-01

**Soundness:** 2 fair
**Presentation:** 2 fair
**Contribution:** 2 fair
**Rating:** 3
**Confidence:** 3

**Summary:**

This paper proposes a method for producing a representation of a protein which can be used for protein function prediction. It is a hierarchical clustering algorithm that iteritively aggregates nodes (Initially nodes are amino acids) into representative nodes.
At each iteration, the algorithm:
1) Generates a new representation for each node. This step uses an attention mechanism and a 3d sphere around the selected node to select the neighborhood.  The radius of the sphere increases at each iteration. This is a called a CRE block and is applied twice per main iteration.
2) A scoring function scores the clusters for each node. The scoring function is a GCN with learnable parameters to weight the features of the mediod node, and features forming the edges between neighboring nodes in the cluster.
The top-N_t scored nodes are retained for the next iteration.
After several iterations, a representation is attained which can be used as input to a fully connected layer that can output protein function predictions.

**Strengths:**

The method produces significant improvements in performance over GearNet and CDConv.
Clustering by a spatial 3D spheriod might capture critical functional regions.

**Weaknesses:**

Attention in the CRE description is called cross attention - but it is more like self attention within group. And it is less like "self attention" as commonly used; Becuase x_n and x_k are never compared or combined (eg. no dot product or qkv-like attention). Instead, the attention learns to weight contributions of features independently in the mediod node and the features in the connected nodes with the single learned w vector (I say the features are weighed independently because the feature vectors x_n and x_k are concatenated). These attention values are then used to weight the contributions of neighbors to the new representation for the mediod node (3). This is ok, but (2) and (3) together amount to a weighted sum of neighbors with weighted features and the weight is not calculated with the relationship between the mediod and edge node in mind. It does work, and probably learns to weight the features for, but it could be better I think.

I cannot work out from the paper how one gets from a protein of variable length to a representation of fixed length that can be used as input to a fully connected layer. If N_t = lower([_w_.N_t-1]) and _w_ is a fixed fraction then this will end in a different number of clusters for different lengths, and there is no mechanism to get to a fixed number of amino acids. Since T is a fixed 4 iterations then the number of remaining amino acids will vary. Please explain this.
If the final selection of nodes is a fixed number, then are the representation vectors of these nodes simply concatenated before the FC? Is there some pooling involved before the FC?

**Questions:**

Why does it say "4 Convs" in the set of CRE blocks of Fig 2 for iteration 1?

In fig 1(c) it states the output is amino acids. And some points in (c) appear to be untouched amino acids. But the text say that only clusters are picked after each iteration. Also, does this figure represent one iteration?